# Reconstructing Boundary Layer Wind Profiles with Machine Learning: A Comparative Study on IGRA Radiosonde Data

## Abstract

Reconstructing atmospheric boundary layer wind profiles is crucial for weather prediction and wind energy applications. We study the task of predicting vertical profiles of the zonal (east–west) and meridional (north–south) wind components from the Integrated Global Radiosonde Archive (IGRA), given geostrophic wind and auxiliary predictors such as season, time of day, temperature difference, and pressure. Instead of relying on fixed-form analytical laws, we propose to treat this as a data-driven learning problem, evaluating CatBoost, TabM, and FT-Transformer against classical baselines such as the power-law profile and Monin–Obukhov similarity theory. On the IGRA dataset, modern ML models achieve reconstruction errors of about 1.5 m/s for both wind components, substantially outperforming analytical parameterizations. These results highlight the potential of data-driven approaches for accurate boundary layer wind modeling.

## 1 Introduction

The atmospheric boundary layer (ABL) is the lowest part of the troposphere where turbulent exchange of momentum, heat, and moisture governs surface–atmosphere interactions. Accurate wind–profile reconstruction in the ABL is essential for short-range weather forecasting, pollutant transport, and wind-energy applications. Radiosondes provide high-resolution vertical soundings, yet their spatial and temporal coverage is sparse, which necessitates predictive models that can infer complete wind profiles from a compact set of predictors.

Modeling wind in the ABL differs fundamentally from modeling in the free troposphere or upper atmosphere. Above the boundary layer, wind profiles are largely governed by large-scale geostrophic balance between Coriolis and pressure-gradient forces, which can be captured with relatively simple dynamical models. Within the ABL, however, turbulent mixing, surface friction, and stratification strongly modify the vertical wind structure. These processes introduce nonlinear and regime-dependent behavior, making boundary-layer profiles far more difficult to approximate with analytical parameterizations. As a result, predictors that are sufficient for upper-air modeling are inadequate near the surface, where local conditions, diurnal cycles, and surface heterogeneity exert a dominant influence. This complexity motivates the use of flexible, data-driven methods that can capture these fine-scale dependencies while remaining grounded in physical predictors such as geostrophic wind, latitude, and boundary-layer depth.

Classical analytical approaches, such as the power-law profile and Monin–Obukhov similarity theory, capture first-order effects of surface roughness and stability but consistently underperform compared to radiosonde observations. These models rely on simplified assumptions and cannot fully represent the diversity of boundary-layer regimes observed in practice.

Recent advances in machine learning provide an alternative, enabling models to capture vertical wind structure directly from data. Transformer-based models (Vaswani et al., 2017), modern tabular architectures such as TabM (Gorishniy et al., 2025), and gradient boosting methods such as CatBoost (Dorogush et al., 2018) have demonstrated strong performance on structured prediction tasks, making them promising candidates for boundary-layer wind reconstruction.

Our contributions are threefold and highlight both methodological and empirical aspects of boundary-layer wind modeling:

1. We cast ABL wind–profile reconstruction as a supervised learning task on the Integrated Global Radiosonde Archive (IGRA) NOAA National Centers for Environmental Information, predicting full vertical profiles $\{u(z), v(z)\}$ from geostrophic wind and auxiliary predictors.

2. We benchmark FT-Transformer, CatBoost, and TabM against classical parameterizations including the power-law profile and Monin–Obukhov similarity theory. Results for Fourier Neural Operators (FNO) (Kovachki et al., 2023) are included in the Appendix for completeness.

3. We provide an upper bound on the effectiveness of ML models by comparing their reconstruction errors to the intrinsic variability of radiosonde profiles, thereby clarifying the extent to which predictive performance is limited by data rather than model capacity.

## 2 PROBLEM FORMULATION

Our objective is to reconstruct the vertical structure of horizontal wind in the atmospheric boundary layer (ABL) from a reduced set of predictors. Unlike in the free troposphere, where wind structure is largely governed by geostrophic balance between pressure-gradient and Coriolis forces, the ABL exhibits regime-dependent variability driven by turbulence, surface friction, and thermal stratification. These processes make boundary-layer profiles far more difficult to approximate with analytical parameterizations, motivating a data-driven approach.

We frame the task as a supervised learning problem. The prediction target is the vertical wind profile inside the ABL, represented by the zonal (east–west) and meridional (north–south) components

$$\{u(z), v(z)\}_{z \in [z_{\min}, H_{\text{ABL}}]},$$

where $H_{\text{ABL}}$ is the diagnosed boundary-layer height from radiosonde measurements and $z_{\min}$ is the surface altitude at the launch site.

Classical analytical parameterizations attempt to reconstruct the wind profile from values specified at the ABL top. For instance, the power-law model assumes that wind speed increases with height according to a fixed exponent:

$$u(z) = u_g \left( \frac{z}{H_{\text{ABL}}} \right)^{\alpha}, \tag{1}$$

where $u_g$ is the geostrophic wind at the ABL top and $\alpha$ is an empirical exponent that depends on surface roughness and stability (typically in the range 0.1–0.3). While such formulations capture first-order trends, they cannot account for the full variability observed in radiosonde data.

In our formulation, the input feature set extends beyond the geostrophic wind and includes:

- the geostrophic wind vector at the top of the ABL, $\vec{g}_w = (u_g, v_g)$;
- latitude, which enters through the Coriolis parameter $f = 2\Omega \sin(\varphi)$;
- thermodynamic predictors such as the temperature difference $\Delta T = T(z_{\min}) - T(H_{\text{ABL}})$ and pressure difference $\Delta P = P(z_{\min}) - P(H_{\text{ABL}})$;
- categorical variables such as season and part of day, which capture systematic diurnal and seasonal variability;
- optional additional predictors (e.g., longitude, station identifiers) considered in extended configurations.

Formally, the learning problem can be written as

$$f : (\vec{g}_w, \Delta T, \Delta P, \text{latitude}, \text{season}, \text{part-of-day}, \dots) \mapsto \{u(z), v(z)\}_{z \in [z_{\min}, H_{\text{ABL}}]}. \tag{2}$$

This formulation provides a unified setting for evaluating classical parameterizations and modern machine learning models, and enables systematic investigation of how different predictor sets affect reconstruction accuracy and generalization across locations.

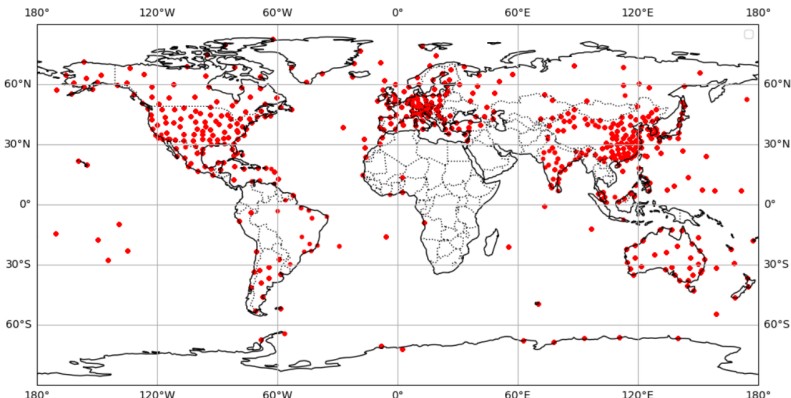

Figure 1: Locations of radiosonde stations from the NOAA Integrated Global Radiosonde Archive used in this study

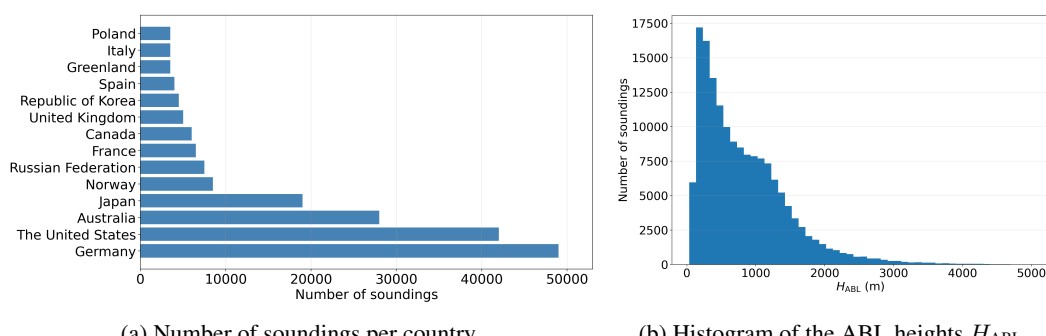

(a) Number of soundings per country.

(b) Histogram of the ABL heights $H_{\mathrm{ABL}}$.

Figure 2: Radiosonde dataset: (a) distribution by country and (b) histogram of the ABL height

## 3 DATASET PREPARATION

### 3.1 DATA OVERVIEW

We use radiosonde profiles from the Integrated Global Radiosonde Archive (IGRA)https://www.ncei.noaa.gov/data/ecmwf-global-upper-air-bufr/archive/. The stations are distributed worldwide (see Figure 1 and 2a). Erroneous soundings (incomplete or physically inconsistent) are removed. For the remaining profiles, the boundary-layer height $H_{\mathrm{ABL}}$ is defined as the smallest root of the following equation (Troen & Mahrt, 1986):

$$\theta(H) = \theta_v(0), \tag{3}$$

where $\theta$ is the potential temperature and $\theta_v$ is the virtual potential temperature at the surface. The values of $H_{\mathrm{ABL}}$ typically range from a few hundred meters to several kilometers, consistent with our computations (see Figure 2b).

To obtain a uniform vertical representation for learning, each interval $[z_{\min}, H_{\mathrm{ABL}}]$ is divided into $N = 50$ sublayers. Profiles are approximated by a Schönberg spline using the normalized height coordinate $\xi = (z - z_{\min})/H_{\mathrm{ABL}} \in [0, 1]$. Soundings with fewer than 20 measurements are discarded. Profiles with $H_{\mathrm{ABL}} < 100$ m, with velocity variation $< 2$ m/s inside the boundary layer, or with extreme velocities $> 50$ m/s are excluded. The dataset after filtering contains 158097 soundings. Figure 3 shows the distribution of launches by month and by time of day.

Before training models, we assess the inherent uncertainty of the reconstruction problem. Specifically, we test the assumption (see Eq. 2) that wind profiles in the ABL can be characterized by the geostrophic wind and a limited set of auxiliary predictors. To quantify this, we partition the dataset into cohorts defined by bins of selected input features (e.g., geostrophic wind components $(u_g, v_g)$, season, part of day, latitude). Within each cohort, profiles are grouped into bins with a

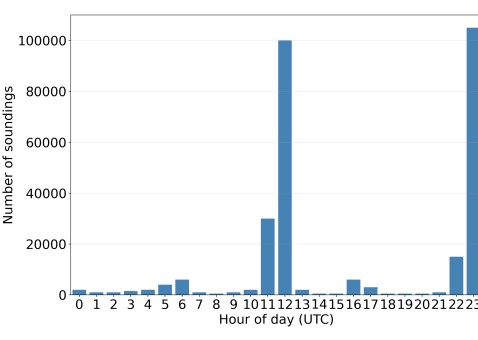

(a) Launch distribution by hour (UTC).

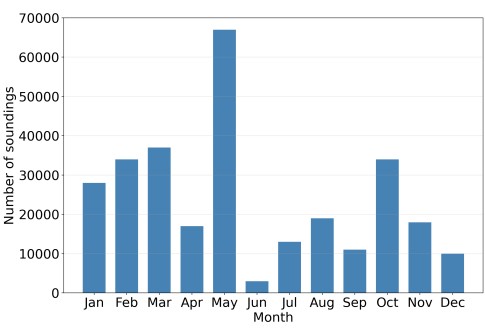

(b) Launch distribution by month.

Figure 3: Radiosonde dataset: (a) distribution of launches by hour of day and (b) distribution by month.

Table 1: Dispersion of wind profiles for different predictor sets. Values represent the average spread (in m/s) of wind speed within cohorts defined by the listed predictors. Lower values indicate that the selected predictors explain a larger fraction of the observed variability.

| Predictor set | Within-cohort dispersion (m/s) |
|---|---|
| $u_g, v_g$ | 3.4 |
| $u_g, v_g$, latitude | 3.2 |
| $u_g, v_g$, latitude, season | 2.7 |
| $u_g, v_g$, latitude, season, part of day | 1.98 |
| $u_g, v_g$, latitude, season, part of day, $H_{\mathrm{ABL}}$ | 1.77 |

resolution of 0.5 m/s in wind speed (or the corresponding step for categorical variables). A smaller bin width would result in very few profiles per bin, while a larger width would artificially inflate the dispersion. The choice of 0.5 m/s balances statistical robustness with a reasonable approximation of within-cohort variability. For every bin, we compute the mean wind profile and its dispersion. The resulting dispersion measures the variability of observed profiles that cannot be explained by the chosen predictors alone. When only geostrophic wind is used, this residual variability is typically around 3 m/s, which sets a lower bound on the achievable reconstruction error for this setting. As additional predictors such as latitude, season, or part of day are included, the within-cohort dispersion decreases, reaching values close to 1.5–2 m/s. This trend demonstrates that richer predictor sets capture more of the underlying variability in the radiosonde data. These values are consistent with Bykov & Gordin (2023), who reported reconstruction errors of about 2 m/s when retrieving the eddy viscosity coefficient from radiosonde measurements.

### 3.2 DESCRIPTION OF CONFIGURATIONS

To evaluate the impact of different predictor sets, we consider two configurations of input features.

**Configuration 1 (canonical).** This configuration represents a minimal set of predictors motivated by physical reasoning. It includes: (i) the geostrophic wind components $(u_g, v_g)$ at the top of the boundary layer, (ii) latitude, which enters through the Coriolis parameter $f$, and (iii) the boundary-layer height $H_{\mathrm{ABL}}$. These predictors reflect the classical assumption that horizontal wind profiles are primarily governed by geostrophic forcing, rotational effects, and boundary-layer depth.

**Configuration 2 (extended).** In addition to the parameters of Configuration 1, this setting incorporates a richer set of predictors aimed at capturing finer dependencies present in radiosonde observations. Specifically, it includes:

- geographical variables (longitude, latitude),
- categorical variables (season, part of day),

- boundary-layer height $H_{\mathrm{ABL}}$,
- surface altitude $z_{\min}$,
- geostrophic wind components $(u_g, v_g)$ at the top of the boundary layer,
- temperature values at surface and ABL top $(T_{z_{\min}}, T_{H_{\mathrm{ABL}}})$,
- pressure values at surface and ABL top $(p_{z_{\min}}, p_{H_{\mathrm{ABL}}})$.

Configuration 2 therefore enables the models to combine physically motivated predictors with additional geographical, categorical, and thermodynamic context, allowing them to account for deviations from idealized analytical formulations.

**Training setups.** In addition to comparing feature configurations, we evaluate two training strategies. *Local models* are trained on subsets of data corresponding to a single country, allowing predictors to specialize in regional atmospheric regimes. *Global models*, in contrast, are trained on the full dataset spanning all radiosonde stations worldwide. This setup leverages the diversity of atmospheric conditions across regions and seasons, but may reduce specialization to local characteristics.

## 4 LEARNING MODELS FOR WIND PROFILE RECONSTRUCTION

### 4.1 FT-TRANSFORMER

The **FT-Transformer** (Gorishniy et al., 2021) is one of the most influential recent deep learning approaches for tabular data. It adapts the Transformer encoder architecture by treating each feature as a token and applying multi-head self-attention across features within a single instance. This design eliminates the need for hand-crafted feature interactions, enabling the model to capture complex, nonlinear dependencies directly from data. By embedding both numerical and categorical predictors into a shared latent space, FT-Transformer provides a flexible and general framework for structured prediction tasks.

For boundary-layer wind profile reconstruction, FT-Transformer is particularly well suited to handle heterogeneous inputs such as geostrophic wind components, temperature and pressure differences, and boundary-layer height $H_{\mathrm{ABL}}$, together with categorical descriptors like season and time of day. Its attention mechanism is advantageous for identifying interactions between physical drivers of boundary-layer dynamics, such as stability, stratification, and geostrophic forcing. We follow the implementation details and hyperparameter recommendations of Gorishniy et al. (2021), tuning embedding dimensions, number of attention heads, depth, and dropout rates to ensure competitive performance. This setup enables a direct comparison with CatBoost and TabM, while highlighting the strengths of feature-wise attention for capturing the vertical wind structure.

### 4.2 CATBOOST

For the tabular formulation, we adopt **CatBoost** (Dorogush et al., 2018) as a representative implementation of gradient boosting over decision trees (GBDT) for mixed-type feature spaces. Its key mechanisms—*ordered boosting* and *ordered target statistics*—mitigate prediction shift due to target leakage and provide stable encodings of categorical features without explicit leakage. CatBoost also employs symmetric (oblivious) trees, in which the same splitting rule is applied at each level, simplifying complexity control and improving reproducibility. Additional mechanisms address missing values and rare categories.

These properties are especially relevant in our setting, where categorical predictors (season, time of day, station or country identifiers) coexist with continuous meteorological variables (geostrophic wind, $\Delta T$, pressure), and the data exhibit strong spatial and temporal structure. As a strong and interpretable baseline, we tune CatBoost hyperparameters under validation schemes that respect temporal and spatial dependencies (e.g., blocked time splits or station-wise partitions), thereby minimizing leakage between training and validation. We optimize tree depth, number of iterations, learning rate, L2 regularization on leaf weights, and early stopping criteria. The loss function (MAE, MSE, or MAPE) is chosen based on the distribution of residuals and robustness to outliers. This protocol establishes CatBoost as a reliable reference model for assessing the benefits of deep learning architectures in our task.

## 4.3 TABM

We also consider **TabM** (Gorishniy et al., 2025), a recent deep learning approach specifically tailored for tabular data. TabM combines the efficiency of compact neural architectures with the robustness of ensembling, yielding a parameter-efficient alternative to large Transformer models on structured inputs. The method is motivated by the observation that strong tabular baselines often rely on ensembles (e.g., GBDT), while single neural networks typically underperform. TabM addresses this gap by employing parameter-sharing strategies across multiple lightweight subnetworks, which are aggregated into an ensemble at inference time. This design balances computational cost and predictive accuracy, while retaining the flexibility of gradient-based optimization.

In the context of boundary-layer wind profile reconstruction, TabM naturally accommodates heterogeneous predictors, including continuous meteorological variables (geostrophic wind, $\Delta T$, pressure, $H_{\mathrm{ABL}}$) and categorical descriptors (season, time of day, station identifier). We tune its hyperparameters following the recommendations of Gorishniy et al. (2025), with special attention to ensemble size, embedding dimensions for categorical variables, learning rate, and dropout regularization. This ensures a fair comparison with CatBoost and FT-Transformer, while highlighting the relative advantages of parameter-efficient neural ensembles on large-scale radiosonde datasets.

## 5 RESULTS

Figure 4 presents the RMSE profiles of reconstructed wind speed within the ABL. The average error is on the order of 1.5 m/s, with a clear vertical structure: errors are largest near the surface and decrease toward the ABL top. This pattern is physically consistent with expectations, since winds aloft are more tightly coupled to the geostrophic flow, while near-surface winds are influenced by turbulence, stratification, and local surface effects not fully captured by the predictors.

Analytical baselines, including the power-law and Monin–Obukhov similarity models, yield RMSE values of approximately 3 m/s, substantially higher than those obtained with machine learning approaches. Among the tested ML architectures, TabM, CatBoost, and FT-Transformer achieve comparable performance, yielding average RMSE values of about 1.3 m/s for local (per-country) models and 1.6 m/s for the global model. Configuration 2 consistently outperforms Configuration 1: adding additional predictors reduces the RMSE from roughly 2.5 m/s to 1.5 m/s. These values align with our preliminary dataset analysis (Table 1), which indicated an intrinsic dispersion of about 1.7 m/s when profiles are conditioned on sufficiently fine bins. This suggests that further improvements are fundamentally limited by the natural variability in radiosonde observations rather than by model capacity.

It is worth noting that analytical models, such as the Åkerblom–Ekman formulation, require specification of a lower boundary condition and knowledge of the eddy viscosity coefficient profile. Neither the boundary condition nor the eddy viscosity coefficients are reliably known, and radiosonde data provide no evidence for a perfectly "slip" boundary at the surface. In contrast, neural models do not rely on such assumptions: their reconstruction error near the surface remains of the same order as in the mid–boundary layer, without requiring explicit boundary conditions.

We next evaluate the ability of the trained models to generalize across geographical locations. To this end, we perform a cross-validation experiment in which each model is trained on data from one location and subsequently validated on radiosonde profiles from all other locations. The resulting cross-validation matrices for Configuration 1 and Configuration 2 using TabM are shown in Figure 5. Each matrix entry reports the maximum RMSE across altitude levels for a given train–test pair of locations.

As expected, the smallest errors are observed along the main diagonal, where training and validation are performed on the same location. When applied to different locations, errors increase, but the degradation remains modest: for Configuration 2, off-diagonal errors exceed the diagonal values by less than 1 m/s. For Configuration 1, the degradation is more pronounced, with increases of up to 1.5 m/s. These results indicate that local atmospheric characteristics shape boundary-layer structure, yet the learned models retain a meaningful degree of transferability across regions, particularly when trained with the richer predictor set of Configuration 2.

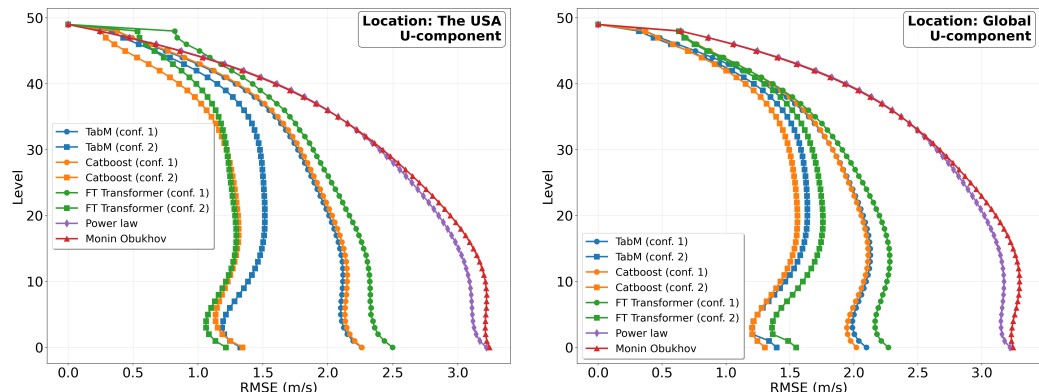

Figure 4: RMSE profiles of wind speed reconstruction in the atmospheric boundary layer. **Left:** results for a representative country-specific (local) model. **Right:** results for the global model trained on all available stations.

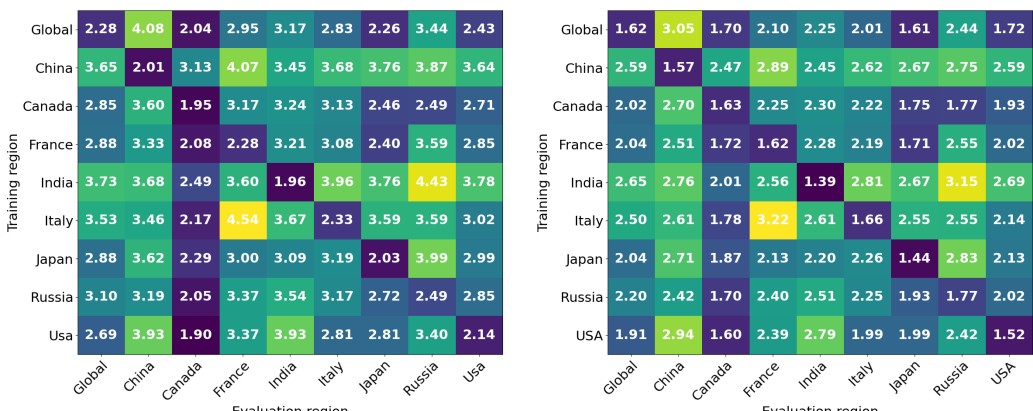

Figure 5: Cross-validation results across geographical locations for TabM model. Each matrix entry shows the maximum RMSE (m/s) across altitude levels for a given train–test pair. **Left:** Configuration 1 (canonical predictors). **Right:** Configuration 2 (extended predictor set).

To further investigate the degree to which models adapt to specific geographical regions, we conduct a focused cross-validation study across the United States. Stations are partitioned into four regional groups, as illustrated in Figure 6. For each group, we train a TabM model using the canonical feature set (Configuration 1) and evaluate it on the remaining groups. The resulting cross-validation matrix is also shown in Figure 6.

As expected, RMSE values along the main diagonal are higher than in the global cross-validation experiment (Figure 5), since training in this case relies on fewer stations and thus less data. However, the off-diagonal errors remain close to the diagonal values, indicating that the learned representations capture region-independent features of boundary-layer wind structure. This suggests that even when trained on geographically restricted subsets, the models retain a largely global character and generalize across different U.S. regions.

## 6 CONCLUSION

We addressed the problem of reconstructing atmospheric boundary-layer wind profiles from radiosonde observations using modern machine learning methods. We benchmarked FT-Transformer, TabM, and CatBoost against classical analytical parameterizations, including the power-law and Monin–Obukhov models.

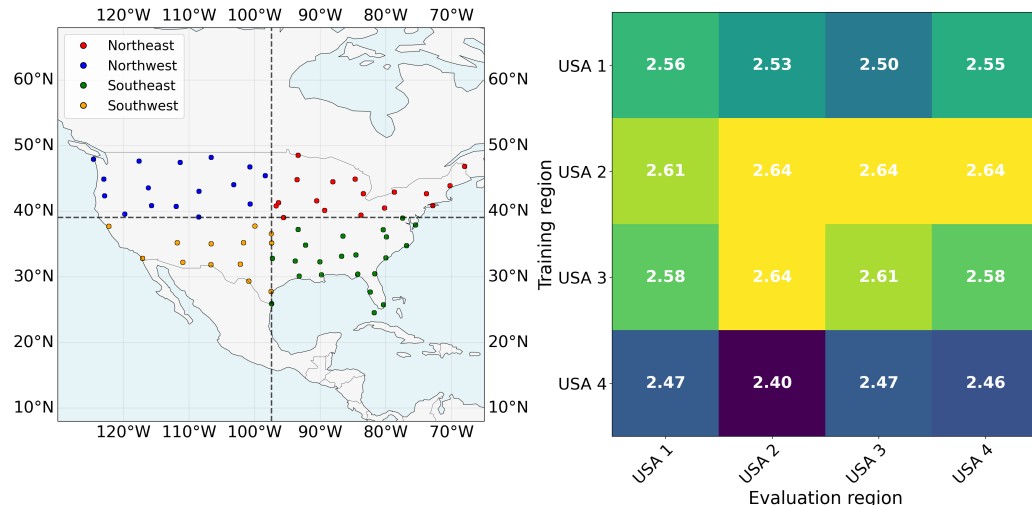

Figure 6: Regional cross-validation across the United States. Left: partition of radiosonde stations into four geographical groups. Right: cross-validation matrix of RMSE values obtained with TabM (Configuration 1). Errors along the main diagonal correspond to models trained and tested within the same region, while off-diagonal entries represent transfer across regions. Comparable performance on and off the diagonal indicates that the learned models generalize beyond their training domain.

Our study leads to three main conclusions. First, machine learning approaches reduce reconstruction errors to about 1.3–1.6 m/s, compared to roughly 3 m/s for analytical baselines, demonstrating a clear advantage of data-driven methods. Second, the predictor set is critical: the extended configuration (Configuration 2) reduces RMSE from about 2.5 m/s to 1.5 m/s, approaching the intrinsic variability of the data ($\approx 1.8$ m/s for a smaller set of predictor features) and thus defining a practical error floor. Third, cross-validation across locations shows that while local models perform best within their training region, global models trained on diverse stations maintain competitive accuracy and transfer effectively across geographical settings.

These results establish machine learning as a reliable tool for ABL wind-profile reconstruction, capable of reaching the accuracy limits imposed by natural variability in radiosonde data. They further highlight the value of combining physically motivated predictors with contextual features to achieve robust generalization. Future work will focus on integrating these models into numerical weather prediction pipelines, applying physics-informed regularization to improve interpretability, extending the methodology to additional boundary-layer variables such as humidity and eventually reducing the uncertainty of global and local surface weather forecasts.

REPRODUCIBILITY STATEMENT

All data used in this study are based on freely available radiosonde measurements from the NOAA National Centers for Environmental Information (NCEI), accessible at `https://www.ncei.noaa.gov/data/ecmwf-global-upper-air-bufr/archive/`.

To ensure reproducibility, we provide the following details:

- **Data preprocessing.** Filtering criteria, boundary-layer height definition, and spline interpolation are described in Section 3.
- **Model configurations.** Hyperparameters and architectural choices for each analytical baseline and machine learning model are given in Appendix.
- **Experiments.** Results in Section 5 are averaged across multiple runs with different random seeds; exact training and evaluation protocols are specified there.
- **Code availability.** An anonymized repository containing preprocessing scripts, model implementations, and experiment configurations will be provided to reviewers upon submission and made public upon acceptance.

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

## A FNO-BASED POST-HOC CORRECTOR

We augment each baseline predictor (CatBoost, FT-Transformer, TabM) with a *post-hoc* one-dimensional Fourier Neural Operator (FNO). The baseline model is kept **frozen**: we first compute its predictions on train/test splits, and then train a lightweight FNO solely on top of these predictions.

Given scaled targets $Y_{\text{train}}^{(t)}$ and $Y_{\text{test}}^{(t)}$, the procedure is as follows:

1. Load the pre-trained baseline and obtain its predictions $\hat{Y}$ for train/test.

2. Spatialize $\hat{Y}$ by tiling along a synthetic 1D axis of fixed length $S = 64$, producing tensors of shape $[B, C, S]$ with $C = N_{\text{targets}}$.

3. Apply the same tiling to the ground-truth targets.

4. Train the FNO to minimize MSE between prediction and target tensors. At inference, predictions are averaged across $S$ to recover tensors of shape $[B, C]$.

For each country we reload the best checkpoint of the baseline and fix all its parameters before generating $\hat{Y}$. The baseline hyperparameters are: CatBoost (3000 iterations, depth 8, MultiRMSE); FT-Transformer ($d = 256$, $h = 16$, $L = 6$, feed-forward $= 512$, dropout 0.2); TabM ($n_{\text{blocks}} = 3$, $d_{\text{block}} = 512$, $k = 32$, PLE embeddings). The compact 1D FNO configuration is kept consistent across baselines: $n_{\text{modes}} = [16]$, hidden channels $= 64$, $C_{\text{in}} = C_{\text{out}} = N_{\text{targets}}$, lifting/projection channels $= 256$, $n_{\text{layers}} = 4$, GELU nonlinearity, linear skip connections, no normalization, and the factorized spectral implementation (implementation=factorized, factorization=tucker, rank=0.5, decomposition=cp). The spatial length is fixed to $S = 64$. Training is performed with AdamW (learning rate $10^{-3}$, weight decay $10^{-4}$), batch size 10k, and up to 1000 epochs with early stopping (patience $= 100$) on validation MSE. A ReduceLROnPlateau scheduler (factor 0.5, patience 20, cooldown 10, minimum LR $10^{-6}$) is applied. We keep the original 80/20 data split, apply z-score scaling to $Y$, and inverse-transform

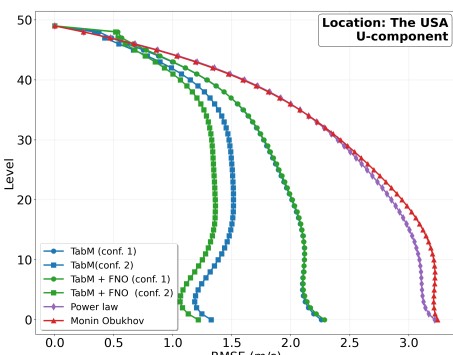

Figure 7: RMSE profiles of wind speed reconstruction in the atmospheric boundary layer using the Fourier Neural Operator (FNO) with Configuration 2 for U.S. stations.

Table 2: Summary of model configurations and compute cost.

| Model | Params | Key Hyperparameters |
|---|---|---|
| FT-Transformer | ∼3–4M | $d = 256$, $L = 6$, $h = 16$, FFN=512 |
| CatBoost | trees: 3000×depth 8 | lr=0.03, GPU, MultiRMSE |
| TabM | multi-M ($\gg$1M) | 3 blocks, $d = 512$, $k = 32$ |

predictions for reporting. For each batch $(X, Y)$ in the FNO dataloader, we forward $X \in \mathbb{R}^{B \times C \times S}$, compute MSE with respect to the tiled $Y \in \mathbb{R}^{B \times C \times S}$, and update the optimizer. We track training/validation losses, schedule the learning rate on validation MSE, checkpoint the best model in memory, and stop early if no improvement is observed. At inference, we average over the synthetic dimension to obtain $\hat{Y}_{\text{FNO}} \in \mathbb{R}^{B \times C}$ and inverse-transform to physical units.

Figure 7 shows the RMSE profiles of wind speed reconstruction in the atmospheric boundary layer obtained with the FNO post-hoc corrector. When applied with the extended predictor set (Configuration 2) for U.S. stations, FNO achieves lower reconstruction errors than TabM, illustrating the potential of operator-learning approaches for this task.

## B   IMPLEMENTATION DETAILS

We trained the following models: **FT-Transformer**, **CatBoost**, and **TabM**. All models used the same preprocessed dataset (categorical features: `season`, `time`; numerical meteorological predictors; target: multi-level wind profiles). The dataset was split 80/20 with stratification by `season` × `time`. Targets were standardized, and inverse scaling was applied for evaluation. Total compute budget is **653 GPU-hours** across all experiments, including per-country models and global runs.

**FT-Transformer.**   Architecture: $d = 256$, 6 encoder layers, 16 attention heads, feed-forward size 512, dropout 0.2, GELU activation, [CLS]-token aggregation. Training used `AdamW` with $lr = 3 \cdot 10^{-4}$, weight decay $10^{-4}$, batch size 30 000, early stopping with patience 250. Model size $\approx$ 3–4M parameters. Training time: **5 h** (global) and $\approx$**1 h** per country.

**CatBoost.**   Configuration: 3000 iterations, depth 8, learning rate 0.03, objective `MultiRMSE`, early stopping with `od_wait=200`, GPU backend. The model consists of 3000 depth-8 trees (up to 256 leaves each). Training time: **2.3 h** (global) and $\approx$**1 h** per country.

**TabM.**   Architecture: 3 blocks of size 512, ensemble size $k = 32$, dropout 0.1, numeric embeddings with piecewise-linear bins (64) and $d_{\text{emb}} = 16$. Optimizer: `AdamW` with $lr = 2 \cdot 10^{-3}$, weight decay $3 \cdot 10^{-4}$, batch size 30 000, early stopping with patience 250. Model size: several million parameters depending on block implementation. Training time: **4 h** (global) and $\approx$**1 h** per country.

