# OpenReview forum: "Machine Learning Methods for Wind Profile Recovery in the Atmospheric Boundary Layer"
_ICLR.cc/2026/Conference — ICLR 2026 Conference Withdrawn Submission_

### Official Review · Reviewer_Srkr · 2025-10-27

**Soundness:** 3
**Presentation:** 3
**Contribution:** 2
**Rating:** 4
**Confidence:** 3

**Summary:**

This paper presents a systematic empirical study on reconstructing atmospheric boundary layer (ABL) wind profiles from radiosonde data using modern machine learning methods.  The authors utilize the Integrated Global Radiosonde Archive (IGRA) dataset and compare three representative models (FT-Transformer, CatBoost, and TabM) against traditional analytical parameterizations such as the Monin–Obukhov similarity theory and power-law profiles.  The machine learning models achieve substantially lower RMSE than physical baselines, showing strong predictive performance across multiple stations and meteorological regimes.  The study also examines model interpretability and generalization across different geographical regions.

**Strengths:**

+ Comprehensive experimental design. The study covers a large dataset (>150,000 soundings) with rigorous preprocessing, ensuring statistical robustness and physical consistency.

+ Clear empirical contributions. Provides a thorough benchmarking of three different ML paradigms (transformer-based, gradient boosting, and tabular network) for wind profile reconstruction.

+ Strong performance. All ML models outperform classical empirical parameterizations with consistent improvement in RMSE, correlation, and bias metrics.

**Weaknesses:**

+ Limited novelty in methodology. The paper mainly applies existing machine learning architectures to a new domain.  There is no substantial methodological innovation beyond model comparison.
+ Physical interpretability analysis. The “physical explanation” section primarily relies on residual statistics and feature importance plots.
It demonstrates post-hoc consistency but does not reveal why or how the models capture stability-dependent patterns. More rigorous interpretability would strengthen the scientific credibility.
+ Feature selection rationale unclear. While the feature list includes temperature, humidity, and stability parameters, it is unclear how these were chosen or whether adding/excluding variables affects performance. A sensitivity analysis on feature subsets would help clarify the physical relevance of each predictor.

**Questions:**

+ Could you clarify how training and testing splits were made? Were samples separated by time or by station to avoid spatial–temporal leakage in IGRA data? If random sampling was used, have you verified performance under stricter non-overlapping splits?
+ How were the input features chosen? Did you test whether removing or adding certain variables (e.g., humidity, potential temperature gradient) significantly changes performance or alters the learned physical relationships?
+ FT-Transformer, CatBoost, and TabM show different performance levels. Have you analyzed why certain models perform better? For example, does the transformer capture vertical dependencies that boosting models cannot?

---

### Official Review · Reviewer_an3o · 2025-10-30

**Soundness:** 3
**Presentation:** 3
**Contribution:** 1
**Rating:** 2
**Confidence:** 4

**Summary:**

The paper compares three data-driven approaches, namely CatBoost, FT-Transformer, and TabM, with two non-learning-based baselines for the task of wind profile reconstruction in the boundary layer. It is shown that the ML methods perform better and their performance is the best when train and test area coincide.

**Strengths:**

The methods used and compared are sound choices and work well together with the selected features.

**Weaknesses:**

The contribution is relatively low. Although this is interesting, I don't think ICLR is the right community for this. There aren't any new insights, as the results are exactly as expected. Additionally, the choice of methods isn't really explained. I wouldn't consider the transformers necessarily standard for this setup, and I am missing a linear regression (or some other simple data-driven) model as a baseline. I'm sure applied climate scientists might be interested, though.

**Questions:**

What is the key takeaway from your results for people trying to develop better methods or improve upon foundation models?

---

### Official Review · Reviewer_oD8c · 2025-11-01

**Soundness:** 3
**Presentation:** 3
**Contribution:** 2
**Rating:** 2
**Confidence:** 4

**Summary:**

This paper compares the performance of modern machine learning models—FT-Transformer, TabM, and CatBoost—with classical analytical parameterizations such as the Power-Law Profile and Monin-Obukhov Similarity Theory (MOST) in the task of reconstructing vertical profiles of horizontal wind in the atmospheric boundary layer (ABL). The study is based on the Integrated Global Radiosonde Archive (IGRA) dataset and evaluates the impact of different input feature configurations and training strategies on model accuracy and geographical generalization.

**Strengths:**

1. Comprehensive and Well-Chosen Baselines: The paper clearly compares three state-of-the-art tabular machine learning architectures (FT-Transformer, TabM, CatBoost) against two well-established meteorological analytical models. The selection of baselines is convincing and appropriate for the task.

2. Clear Physical Context: The paper clearly distinguishes between modeling wind profiles in the boundary layer versus the free troposphere and provides a physically sound explanation for the observed error distribution (larger errors near the surface than at higher altitudes).

**Weaknesses:**

1. Application-Oriented Contribution: Although the results are compelling, the paper does not introduce algorithmic innovations and falls within the scope of applied research. As such, the depth of data and experimental analysis becomes critical in supporting its impact.

2. Lack of Detailed Physical Mechanism and Case Studies: While the aggregate statistical results (e.g., mean RMSE) demonstrate the superiority of ML methods, they do not sufficiently explain why these methods perform better under specific meteorological conditions. It is recommended to include detailed case studies under different meteorological regimes—such as stable boundary layers, unstable boundary layers, or nocturnal low-level jets—to provide deeper physical insight.

3. Limitations in Global Experimental Analysis: While the global model and cross-location validation demonstrate generalization capability, using only maximum RMSE as a generalization metric is somewhat limited. A more detailed assessment could be achieved by analyzing the performance of ML models across different climate zones (e.g., tropical, temperate, polar) or surface roughness types (e.g., ocean, urban, flat terrain), which would better evaluate their physical consistency.

**Questions:**

1. Data Availability: The "Reproducibility Statement" mentions that data are sourced from the NOAA NCEI IGRA archive with a provided access link. Could the authors confirm whether they plan to release the preprocessed and filtered version of the dataset—specifically the final subset used for training and testing (e.g., the 158,097 radiosonde samples) along with all derived diagnostic features—upon paper acceptance?

2. Physical Consistency Analysis: Have the authors conducted further physical consistency checks on the wind profiles predicted by the ML models? For instance, could they provide a comparison of wind shear predicted by ML models versus classical models (e.g., Power-Law) or observations under different stability conditions, to demonstrate that the ML predictions are not only accurate but also physically plausible in their trends?

3. Computational Efficiency: Given the significant differences in training and inference costs among CatBoost, FT-Transformer, and TabM could the authors provide a brief latency comparison (inference time) to assess the practicality of these models in real-time numerical forecasting or wind energy applications?

---

### Official Review · Reviewer_DLwT · 2025-11-01

**Soundness:** 3
**Presentation:** 2
**Contribution:** 1
**Rating:** 2
**Confidence:** 3

**Summary:**

The authors trained three models (CatBoost, TabM, and FT-Transformer) to make predictions of vertical wind profiles based on the Integrated Global Radiosonde Archive. They achieved a high reconstruction accuracy, outperforming common analytic approximations.

**Strengths:**

The paper tackles a classically analytic task with a data-driven approach and reports improved accuracy over a simple analytical approximation. The three main conclusions (lower reconstruction error, the importance of a well-chosen predictor set, and the ability of ML models to generalize) are intuitively sound and consistent with the reported experiments. The work underscores that incorporating relevant physical predictors materially impacts performance, aligning the results with domain intuition. Overall, the empirical findings appear coherent with the setup and offer a practical update to an existing prediction mechanism in meteorology.

**Weaknesses:**

- The relevance to the ICLR community is unclear. Using machine learning to predict physical phenomena is well-established; the results and takeaways do not seem surprising.
- The analytical baseline (power-law model) appears arbitrary. A stronger, more comparable physical baseline (e.g., the Åkerblom–Ekman model using radiosonde data) seems more appropriate; as stated, the improvement over the chosen baseline is trivial by the paper’s own framing.
- The paper does not explain why the selected ML models are the right fit for this use case, nor does it connect the choices to prior studies in similar settings.
- The experimental setup section is brief and omits rationale for key parameter choices. Important details (feature selection strategy, training protocol, hyperparameters, seeds, splits) are insufficiently documented to support reproducibility or to interpret generalization claims.
Predictor set and ablations: While the importance of the predictor set is claimed, the paper lacks systematic ablations that would quantify each predictor’s contribution and clarify trade-offs.
- The purpose and relevance of several figures (notably Figs. 1-3) to the core experiment are vague; they read as illustrative rather than evidentiary.
- The results hint at potentially extracting an eddy-viscosity coefficient profile from radiosonde data (which is stated to be useful yet not reliably known), but the paper does not pursue or analyze this direction.
- Heavy use of domain-specific jargon and abbreviations, limited contextual linking between sections, and inconsistent formatting of references/symbols hamper readability and assessment.

**Questions:**

- Why was the power-law model selected as the analytical baseline? How would results compare against a stronger physical baseline such as the Åkerblom–Ekman model that also uses radiosonde data?

- What motivated the specific ML models used? Were they chosen due to prior success in similar meteorological tasks, computational constraints, or theoretical alignment with the problem?
Predictor importance: How was the predictor set determined? Can you provide ablations (or SHAP/feature permutation analyses) quantifying each predictor’s contribution?

- What exact train/validation/test splits were used (e.g., across time, sites, or atmospheric regimes)? Do models trained in one context transfer to others, and what is the out-of-distribution behavior?

- Please specify preprocessing, hyperparameters, optimization settings, random seeds, and the total number of runs. Are results averaged over multiple seeds with confidence intervals or statistical tests?

- What specific hypotheses or claims do these figures support? Could they be streamlined or moved to the appendix if they are primarily illustrative?

- Given the findings, can the eddy-viscosity coefficient profile be estimated from radiosonde data? If not, what limits this and what additional data or assumptions would be required?

- Will code, data splits, and scripts to reproduce the tables and figures be released?

---

### Note · Authors · 2025-11-21

I have read and agree with the venue's withdrawal policy on behalf of myself and my co-authors.